# The Value of Posterior Cervical Angle as a Predictor of Vaginal Delivery: A Preliminary Study

**DOI:** 10.3390/diagnostics11111977

**Published:** 2021-10-25

**Authors:** Eun-Ju Kim, Ji-Man Heo, Ho-Yeon Kim, Ki-Hoon Ahn, Geum-Joon Cho, Soon-Cheol Hong, Min-Jeong Oh, Nak-Woo Lee, Hai-Joong Kim

**Affiliations:** Department of Obstetrics and Gynecology, Korea University School of Medicine, Seoul 02841, Korea; zzuya90@naver.com (E.-J.K.); heojiman.md@gmail.com (J.-M.H.); akh1220@korea.ac.kr (K.-H.A.); geumjoon@korea.ac.kr (G.-J.C.); novak082@korea.ac.kr (S.-C.H.); mjohmd@korea.ac.kr (M.-J.O.); haijkim@korea.ac.kr (H.-J.K.)

**Keywords:** singleton, posterior cervical angle, vaginal delivery

## Abstract

Accurate prediction of failure to progress and rapid decision making regarding the mode of delivery can improve pregnancy outcomes. We examined the value of sonographic cervical markers in the prediction of successful vaginal delivery beyond 34 weeks of gestation. A retrospective chart review was carried out. Medical information of singleton gestations delivered at a single center from 1 July 2019 to 30 August 2020 was collected. Transvaginal sonographic records of cervical length, anterior and posterior cervical angles, and cervical dilatation were obtained and re-measured. The value of these markers and clinical characteristics of mother and baby on vaginal delivery were investigated and compared to women who underwent cesarean section. A total of 90 women met the inclusion criteria. The rate of vaginal delivery was 75.6%. There were no differences found in terms of maternal age, rate of abortion, induction of labor, premature rupture of membranes, preterm labor, hypertension, diabetes, cervical length, and neonatal sex and weight. The prediction of vaginal delivery was provided by parity, maternal body mass index, and posterior cervical angle. The area under the receiver operating characteristic curve for prediction of vaginal delivery was 0.667 (95% CI 0.581–0.864, *p* = 0.017) for the posterior cervical angle, with a cutoff of 96.5°. Regression analysis revealed a posterior cervical angle ≥96.5° in the prediction of vaginal delivery (adjusted odds ratio: 6.24; 95% confidence interval: 1.925–20.230, *p* = 0.002). Posterior cervical angle ≥96.5° is associated with successful vaginal delivery. It is simple and easy to measure and can be useful in determining the mode of delivery.

## 1. Introduction

“Failure to progress” is a leading indication for primary cesarean section [1]. In a recent study, failure to progress accounted for 68% of unplanned cesarean sections [2]. Accurate prediction of failure to progress and rapid decision making regarding the mode of delivery can improve pregnancy outcomes as well as reducing unnecessary time and effort in vaginal delivery [3]. Various predictors of successful labor have been studied. The Bishop score has been widely used to assess the condition of the cervix, thus predicting vaginal or cesarean section [4]. However, the Bishop score is subjective and has a poor predictive value [5].

To overcome the disadvantages of the Bishop score and discover objective predictors, various methods using ultrasound have been studied [6]. Prediction models for successful delivery through ultrasound include cervical length, angle of progression (AOP), fetal head–perineum distance (HPD), fetal head–symphysis pubis distance (HSD), fetal head position before labor, sono-elastography, and shear wave velocity, which allow the assessment of cervical stiffness in pregnancy [7]. Although studies on cervical length as a predictor of delivery mode have been conducted widely, the results are controversial [8,9,10,11,12]. Sono-elastography has been introduced recently and proven to be useful for predicting the outcome of induced labor; however, this technique it is unsuitable for general use because of expensive equipment and the requirement of high-level skills. Other sonographic parameters such as intrapartum ultrasonography measurements were used to assess fetal head descent as well as efficiency during labor. However, there is no sonographic prediction method clearly presented as a golden standard.

The Uterocervical angle (UCA), which is defined as the angle between the anterior and posterior uterine wall and the cervical canal, represents a novel ultrasonographic marker. Cannie et al., who supported the efficacy of the Arabin pessary in preventing preterm birth, revealed that altering the UCA into a more acute angle via pessary insertion has been confirmed by magnetic resonance imaging [13].

In recent decades, several studies have investigated the potential value of UCA, especially the posterior cervical angle (PCA), as a mechanical barrier for the prediction of preterm birth [9,14], with few studies reporting the influence of PCA on labor progress [15].

The objective of this study was to evaluate the value of UCA measurement in the prediction of successful vaginal birth before the initiation of labor beyond 34 weeks of gestation.

## 2. Materials and Methods

### 2.1. Patient Characteristics

We included women who had singleton delivery beyond 34 weeks of gestation with a cephalic presentation at our hospital from 1 July 2019 to 30 August 2020. The evaluated clinical data included demographic information, labor and delivery data, the presence of co-morbidities (hypertension and diabetes), obstetric complications (preterm labor and premature rupture of membranes), neonatal sex, and birth weight. Failure to progress was diagnosed under the standard accepted definition [15]. Women with multiple pregnancies, preterm birth less than 34 weeks of gestation, cesarean section indications other than failure to progress, persistent occiput posterior position, and no sonographic record were excluded. This study was approved by the Institutional Review Boards (2021AS0006) and Informed consent was waived by the Institutional Review Board.

### 2.2. Ultrasound Examinations

Ultrasound examinations were performed using an ultrasound machine (Samsung HERA W10, Seoul, Korea; GE Healthcare Voluson E10, Tiefenbach, Austria) with a 5- to 9- MHz transvaginal transducer. All ultrasound examinations were performed by obstetricians who have more than 2 years of experience in conducting transvaginal ultrasound examinations. Women who were eligible for our study were selected and ultrasound images were reviewed blindly by obstetricians. The timing of the ultrasound examination was at the third trimester before labor.

Women were placed in the supine position with hips flexed and abducted, and knees flexed, and with an empty bladder. The vaginal transducer was covered with a single-use sterile vinyl. It was ensured that the ultrasound images contained the internal and external parts of the cervix and the outermost part of the fetal presenting part. The cervical length or distance between the internal cervical orifice and the external cervical orifice was measured, and the anterior and posterior cervical angles were measured by drawing two lines that converged in the internal cervix of the orifice, according to the methodology described by Rane et al. [6] (Figure 1). In cases wherein the cervix shows funneling and curve, the angle was evaluated at the convergence of the line measuring the cervical length and posterior uterine wall (Figure 1). Several images were recorded at the time of examination, and the best image was chosen for review. Length and angles were measured thrice, and the average was recorded.

### 2.3. Statistical Analysis

Normality test was performed using the Shapiro–Wilk test for continuous variables. When the distribution was normal, the data were expressed as mean values and standard deviation (SD), while medians and interquartile ranges were used when the data were not normally distributed. Categorical variables and continuous values were estimated using appropriate statistical tests for comparison (i.e., Mann–Whitney U-test, chi-square test, and Fisher’s exact test).

The significance level was set at 0.05. The optimal cutoff for PCA was calculated using receiver operating characteristic (ROC) curve analysis. Multiple logistic regression was used to predict successful vaginal delivery. The confounding factors were maternal BMI, parity, and cervical length. All statistical analyses were performed using SPSS version 19.0 (SPSS for Windows; SPSS Inc., Chicago, IL, USA).

## 3. Results

During the study period, there were 331,510 live births in South Korea. In total, 385 women delivered at our institution and 90 women were selected for analysis (Figure 2). Among these, 58.8% (53/90) were primiparous. Successful vaginal delivery occurred in 68 women (75.6%). No significant difference was found between the two groups in terms of maternal age, gestational age, the rate of abortion, induction of labor, premature rupture of membranes, preterm labor, hypertension, diabetes, neonatal sex, and birth weight (Table 1). The rate of nulliparity and assisted reproductive technology was significantly lower in the vaginal delivery group. The amniotic fluid index and anterior cervical angle were similar between the two groups. Cervical length was significantly shorter in the vaginal delivery group and PCA was significantly wider in the vaginal delivery group (Table 2).

BMI, body mass index; PROM, premature rupture of membranes; PPROM, preterm premature rupture of membranes; ART, assisted reproductive technology.

The optimal PCA cut-off value, which was calculated from the ROC curve, was 96.5° (*p* = 0.017) with a sensitivity of 73.5% and specificity of 63.6% (Figure 3). In primiparity, the cut-off value was 97.2° (*p* = 0.026) with a sensitivity of 74.4% and specificity of 65%.

Logistic regression analyses to predict successful vaginal delivery are presented in Table 2 and Table 3. Higher maternal BMI, multiparity, and PCA greater than 96.5° were independent factors associated with successful vaginal delivery. In the nulliparity group, a PCA greater than 97.2° was the only independent factor associated with successful vaginal delivery.

## 4. Discussion

We performed a retrospective cohort study to investigate the value of UCA in the prediction of successful vaginal birth before the initiation of labor beyond 34 weeks of gestation. Our study showed that a wider PCA is associated with higher chances of successful vaginal delivery regardless of BMI and parity. Additionally, PCA is an independent factor that is related to successful vaginal delivery in nulliparity. In contrast, ACA measurement was not valuable in the prediction of successful vaginal birth, because pressure is more often applied to the posterior wall than the anterior wall. The cervix is composed mainly of collagen and should be able to withstand mechanical pressure from surrounding pelvic structures, especially the enlarging uterus. The remodeling of collagen fibers in the cervix varies individually during pregnancy. In addition, collagen fiber orientation and dispersion vary according to anatomic characteristics [16,17]. These micro-environmental factors contribute to cervical stiffness, which reflects PCA.

PCA has been studied by many authors to predict successful labor induction. Keepanasseril et al. suggested that a PCA of at least 100° for the prediction of successful induction of labor in nulliparous women is accompanied by a sensitivity and specificity of 65% and 72%, respectively [18]. Al-Adwy et al. also supported that a PCA of more than 99.5° yielded the best accuracy in predicting the successful induction of labor [19]. These studies have shown that PCA assessed by transvaginal ultrasound is a better predictor than the Bishop score for predicting vaginal delivery. Our study results are compatible with those of previous reports. However, few studies to date have evaluated the role of PCA measurement in outcome prediction regardless of induction. In this study, we evaluated the PCA as an independent predictive factor of successful vaginal delivery regardless of induction of labor, maternal BMI, and parity beyond 34 weeks gestational age. Considering that the measurements were obtained during the antenatal period in the third trimester, PCA might be a useful predictor of successful vaginal delivery before labor.

The relationship between the cervical canal, internal os, and uterine wall might be one of the essential factors related to labor progression. A wider PCA leads to easier passage of the baby through the birth canal, while a narrower PCA would lead to less direct force on the internal os, followed by difficult labor progression. In addition, increased PCA represents a more anterior cervix position. It is correlated with the accurate position of the cervix, which is also known as one of the Bishop score elements. Eggebo et al. suggested that manual assessment of the cervical position may possibly be replaced by ultrasound measurements [20]. To overcome the subjectivity and low predictive value of the Bishop score, many studies are working on developing ultrasound measurement as an alternative [6,15,19,20]. Our results provide evidence to support this stance.

Our findings suggest that individual PCA reference models depending on maternal characteristics may aid in the standardization of delivery mode prediction. In our study, maternal BMI and parity were considered in developing the model as previous studies report that these factors strongly influence the mode of delivery [21,22]. However, our results show that in nulliparous women, PCA was the only factor associated with successful vaginal delivery, and the best cut-off point was 97.2°. Therefore, PCA can be used as a more independent predictor, especially in nulliparous women; however, this should be interpreted with caution due to the small number of nulliparous patients in our cohort. A prediction model that presents individual cut-off criteria according to maternal BMI or parity may be needed to further improve the predictive value of the PCA.

This study has its limitations. First, for a retrospective study, the sample size is small; however, it is meaningful as a pilot study that affirms the significance of PCA and provides a foundation for further prospective research. Second, differences in ultrasound measurement methods or records between examiners might have occurred, and there may be differences in results according to induction delivery methods; however, these differences may also possibly reflect what occurs in actual clinical settings. Third, other factors that can affect PCA were not explored. Congenital uterine version and flexion are correlated with this angle [23]. Retro-version and flexion compared with antero-version and flexion may reduce the physiologic stress on the cervix and reduce mechanical forces affecting cervical dilatation. Physiologic changes, particularly collagen fiber redistribution and orientation in the upper cervix during pregnancy, may also be related to cervical flexibility and stiffness, which contribute to the UCA [17]. Further studies that evaluate the impact of these factors on the UCA will be necessary to clarify this relationship. Cervical repair is accomplished progressively according to gestational age. Therefore, sonographic findings may differ depending on gestational age [24]. Multiple logistic regression analysis was adjusted for pregnancy duration to eliminate the effect of gestational age. The final limitation is that the hospital in which the study was conducted is a tertiary hospital, so it is an environment with relatively many high-risk mothers. Because of this, there may be relatively more cesarean deliveries than the general population. Therefore, further studies involving more patients with uncomplicated pregnancies are needed.

## 5. Conclusions

Prediction of successful vaginal delivery can be provided by pre-labor PCA ultrasound measurement in our cohort. We suggest considering this parameter combined with other factors when deciding the mode of delivery. Future prospective trials are needed to confirm our findings and to define the role of PCA as a screening method to determine the mode of delivery.

## Figures and Tables

**Figure 1 diagnostics-11-01977-f001:**
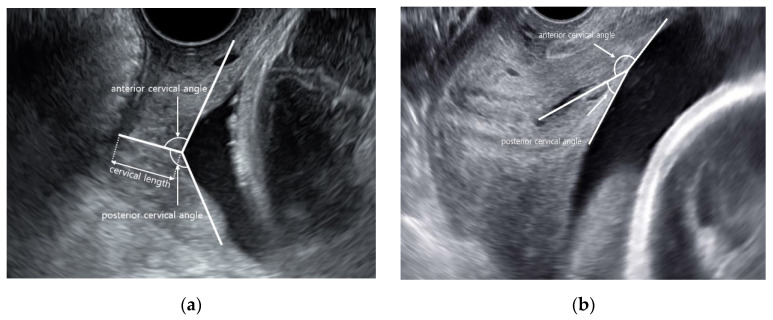
Transvaginal ultrasound cervical measurements. Cervical length, angle between anterior and posterior uterine wall, and internal cervical os in straight cervix (**a**) and curved cervix (**b**).

**Figure 2 diagnostics-11-01977-f002:**
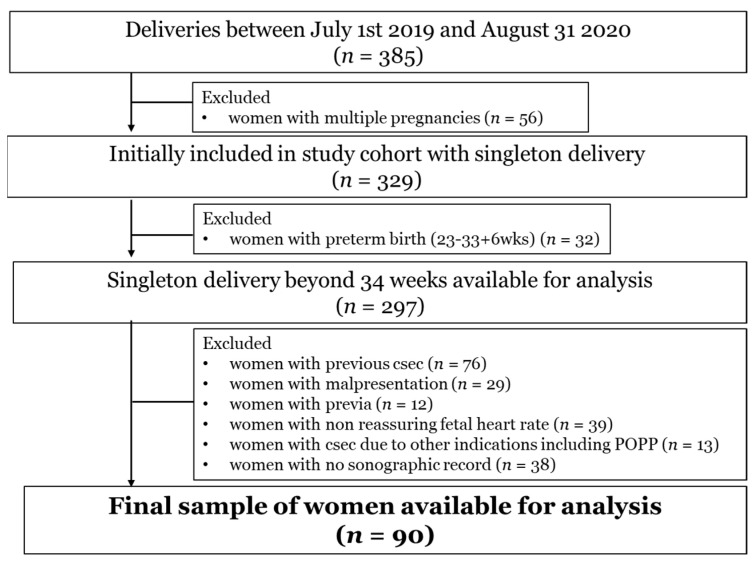
Flowchart of study population.

**Figure 3 diagnostics-11-01977-f003:**
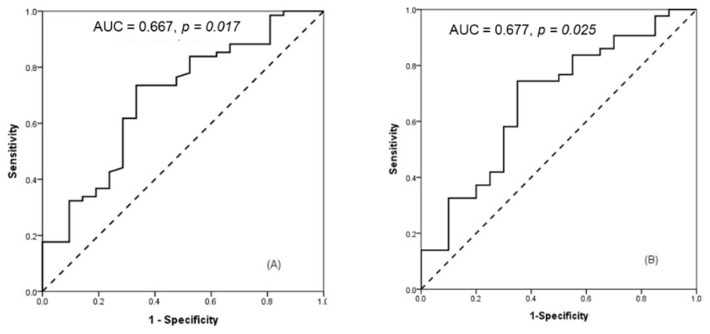
Receiver operating curve for posterior cervical angle (**A**) and (**B**) in primiparity.

**Table 1 diagnostics-11-01977-t001:** Patient characteristics according to mode of delivery.

	Vaginal Delivery (*n* = 68)	Cesarean Section (*n* = 22)	*p*-Value
Age (years)	32.6 ± 4.5	33.1 ± 5.8	0.703
Gestational age (days)	264.9 ± 14.5	270.6 ± 11.8	0.101
Multiparity (%)	36.8	4.5	0.004
Abortion (%)	30.8	50	0.127
BMI (kg/ht2)	26.3(19.5–43.3)	29.2(20.6–41.8)	0.033
Epidural anesthesia (%)	58.8	40.9	0.218
Induction of labor (%)	27.9	50	0.062
PROM or PPROM (%)	25	22.7	0.829
Preterm labor (%)	13.2	4.5	0.440
ART (%)	0	18.2	0.003
Hypertension (%)	5.9	9.1	0.632
Diabetes (%)	11.8	18.2	0.478
Neonatal male gender (%)	52.9	40.9	0.328
Birth weight (kg)	3.31 ± 0.40	3.46 ± 0.41	0.07

BMI, body mass index; PROM, premature rupture of membranes; PPROM, preterm premature rupture of membranes; ART, assisted reproductive technology.

**Table 2 diagnostics-11-01977-t002:** Ultrasound results.

	Vaginal Delivery (*n* = 68)	Cesarean Section (*n* = 22)	*p*-Value
AFI (cm)	11.7 (9–14.7)	12.6 (10.3–15)	0.33
Cervical length (mm)	25.5 ± 1.26	32.4 ± 1.37	0.034
Anterior cervical angle (°)	115.2 ± 21.4	117.4 ± 33.2	0.619
Posterior cervical angle (°)	106.8 ± 26.9	88.7 ± 30.4	0.0096

AFI, amniotic fluid index.

**Table 3 diagnostics-11-01977-t003:** Logistic regression analysis for likelihood of vaginal delivery.

	OR	*p* = Value	aORs (95% CI) *	*p* = Value
Maternal BMI	0.883 (0.800, 0.976)	0.015	0.861 (0.764, 0.971)	0.013
Multiparity	12.209 (1.547, 96.348)	0.018	20.393 (1.818, 228.8)	0.015
Cervical length	0.652 (0.434, 0.978)	0.039	0.669 (0.411–1.09)	0.107
Posterior cervical angle ≥ 96.5°	4.012 (1.467, 10.975)	0.007	5.342 (1.553, 18.369)	0.008

BMI, body mass index, * adjusted for maternal BMI, multiparity, cervical length, posterior cervical length.

## Data Availability

Data are available on reasonable request.

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
