# Peer review of "The Value of Posterior Cervical Angle as a Predictor of Vaginal Delivery: A Preliminary Study"

_diagnostics, 2021, doi:10.3390/diagnostics11111977_

Round 1

Reviewer 1 Report

This is an interesting study providing conclusions of considerable practical usefulness. The Authors correctly state in the discussion that the small number of patients is a limitation of this study. Therefore I suggest to add to the title "preliminary study", the term they use themselves in the text. 

Author Response

We really appreciate the comment on our paper. We added ‘preliminary study’ in the title as suggested.

Reviewer 2 Report

REVIEW

Manuscript ID: diagnostics-1377255

Manuscript details:

Journal: Diagnostics

Title: The value of posterior cervical angle as a predictor of vaginaldelivery

Authors: Eun Ju Kim, Ji Man Heo, Ho Yeon Kim *, Ki-Hoon Ahn, Geum Joon Cho,

Soon Cheol Hong, Min-Jeong Oh, Nak Woo Lee *, Hai-Joong Kim Submitted to

section: Pathology and Molecular Diagnostics,

The Bishop score is a scoring system for the cervix to help predict successful induction of labor. It based on manual investigation, therefore interpretation need long time practice and it is a subjective scoring system. While Bishop score has been found to be widely used for predicting vaginal delivery with sensitivity around 65% as well as a positive predictive value 70%, it has poor specificity and negative predictive value.

The manual examination is its simplicity, modern obstetricians need new, more objective and accurate methods for cervical assessment.

This study is very useful to apply the US as a simple, non invasive method to predict management of labor.

Title:

I should add the terminus ’pilot study’, because the number of cases is very small.

Introduction:

The authors have to present the new US methods for cervical investigation e.g. ellastography.

M&M

Give data of all birth number in Korea, birth number in Department of Obstetrics and Gynecology, Korea University School of Medicine, Seoul, Korea in the study period.

There is 3,945,159 live births between 2009 and 2017 in South Korea ( Obstet Gynecol Sci 2020; 63(5): 623-630) in 5 centers. It means cc 10 000 live births a year per center. The publication included the center Department of Obstetrics and Gynecology, Korea University Medicine, Seoul, Korea, as well.

Ref.:

       1-Obstet Gynecol Sci 2020; 63(5): 623-630)

      2-https://www.macrotrends.net/countries/KOR/south-korea/birth-rate

“During the study period, 385 women delivered at our institution”-result section-it is extremely low number. Is it correct?

It is very important data the ratio of singleton pregnancy and the ratio of cephalic position in your center during study periods.

Give the data of recruitment success.

“ Cervical angle” Ultrasound examination is time consuming. How long was the average time to declare the cervical angles?

The number of cases are very small, it is only a pilot study.

Table 2

Give the exact gestational age at diagnosis of AFI, CxL, Ant C angle, Post C angle.

Give data of intra/inter observer error.

References

There is numbering problem from ref 8.
